# Graphene/Ge Photoconductive Position-Sensitive Detectors Based on the Charge Injection Effect

**DOI:** 10.3390/nano13020322

**Published:** 2023-01-12

**Authors:** Genglin Li, Jintao Fu, Feiying Sun, Changbin Nie, Jun Wu

**Affiliations:** 1School of Optoelectronic Engineering, Chongqing University of Posts and Telecommunications, Chongqing 400065, China; 2Chongqing School, University of Chinese Academy of Sciences, Chongqing 400714, China; 3Chongqing Institute of Green and Intelligent Technology, Chinese Academy of Sciences, Chongqing 400714, China; 4University of Chinese Academy of Sciences, Beijing 100049, China

**Keywords:** position-sensitive detectors, graphene, photoconductive, charge injection effect, heterojunction

## Abstract

Position-sensitive detectors (PSDs) are of great significance to optical communication, automatic alignment, and dislocation detection domains, by precisely obtaining the position information of infrared light spots which are invisible to human eyes. Herein, a kind of PSD based on graphene/germanium (Ge) heterojunction architecture is proposed and demonstrated, which exhibits amplified signals by unitizing the charge injection effect. Driven by the graphene/Ge heterojunction, a large number of photogenerated carriers diffuse from the incident position of the light spot and subsequently inject into graphene, which ultimately generates a photoresponse with high efficiency. The experimental results show that the device can exhibit a fast response speed of 3 μs, a high responsivity of ~40 A/W, and a detection distance of 3000 μm at the 1550 nm band, which hints that the graphene/Ge heterojunction can be used as an efficient platform for near-infrared light spot position sensing.

## 1. Introduction

PSDs are optoelectronic position sensors that can accurately detect the light spot position, distance, incident angle, and other related physical parameters [1,2,3,4]. The lateral photovoltaic effect (LPE) and the photoconductive effect (PCE) have been investigated and utilized to design high-performance PSDs [5]. For LPE devices, the photogenerated electron-hole pairs are separated by the built-in electric field, and the electrons and holes will drift to the surface transport layer and the bulk semiconductor, respectively. The separated electrons and holes weaken the built-in field near the light spot, and thus a potential difference will be formed which pushes electrons (or holes) to move towards the electrodes. Because the potential difference is related to the distance between the light spot and the electrode, the position of the light spot can be calculated by measuring the potential differences between each electrode and the light spot. As for PCE devices, when the semiconductor material absorbs photons under illumination, the carrier concentration dramatically increases at the incident position of the light spot. Subsequently, these massive free carriers diffuse around and increase the conductivity of the material. Thus, the light spot position can be calculated according to the variation of conductivity in the photoconductive devices.

The silicon-based PIN [6] and Schottky [7] PSDs exhibit a strong signal-to-noise ratio, high resolution, and low power consumption in the visible region. However, limited by the bandgap of silicon (1.1 eV), the response range is generally within 1130 nm [8]. With the increasing development of optical communication, PSDs used for light spot position sensing and optical path alignment at 1550 nm are highly desired. Thus, it becomes increasingly important to fabricate a device with a wide response spectrum. InGaAs and Ge are primary materials with an optical response covering 1550 nm. However, optoelectronic devices based on InGaAs material often require low-temperature environments to improve the quantum efficiency and signal-to-noise ratio. Furthermore, InGaAs devices also have the shortcomings of high cost and complex preparation [9]. In contrast, Ge has a suitably narrow bandgap and exhibits a maximum absorption rate at 1550 nm, which is suitable for fabricating near-infrared PSDs. However, the Ge-based PSDs [10] usually face similar challenges as the Si-based pin and Schottky devices have, that is limited gain and detection distance [11,12]. Moreover, the doping process will usually introduce impurities and defects for the LPE devices, resulting in a degraded response speed [13]. Thus, new device structures should be proposed to achieve sensitive PSD by integrating with other materials and adopting a suitable fabrication process.

Two-dimensional materials are those in which electrons can move freely in only two dimensions, including graphene, black phosphorus, and transition metal dichalcogenides [14]. Two-dimensional materials have exhibited unique optical and electrical properties, including high carrier mobility, tunable band structure, and strong light-matter interaction, which make them widely used in the preparation of optoelectronic devices. The photodetectors based on two-dimensional materials have shown excellent performance, including fast speed, high responsivity, and broadband detection [15,16,17]. Among them, graphene has attracted intense attention due to its excellent properties, such as mechanical flexibility, high mobility, and layer-tunable band structures [18]. In particular, the surface of graphene is naturally free of dangling bonds and graphene can be transferred onto any semiconductor without the lattice-mismatch issue [19,20]. Thus, high-quality heterojunctions can be obtained without introducing the doping process. Moreover, graphene-based hybrid photoconductors and phototransistors have been demonstrated to have ultrahigh photogain [21,22,23]. Therefore, it is of great value to improve the performance of PSDs [24,25,26] by combining graphene with traditional semiconductors.

In this paper, a photoconductive type of PSD based on graphene/Ge heterojunction by using the charge injection effect is presented. Under near-infrared illumination, the photogenerated carriers in Ge diffuse laterally to the graphene/Ge heterojunction region and then inject into graphene. During this process, the graphene/Ge heterojunction plays a critical role in efficiently collecting the photogenerated carriers. As a result, the graphene/Ge photoconductive PSD shows a responsivity of ~40 A/W and a response speed of 3 μs/1.9 μs. More importantly, the device shows a dependent photocurrent variation with the position change of the 1550 nm light spot, and its working distance can reach 3000 μm.

## 2. Device Structure and Working Mechanism

Figure 1a shows the schematic view of the graphene/Ge photoconductive PSD. The intrinsic Ge substrate acts as a near-infrared absorber, while graphene serves as the carrier transport channel. Two gold electrodes directly connect with graphene for signal readout. The working mechanism can be understood as follows. When a 1550 nm laser illuminates the device, a large number of photogenerated carriers are generated in Ge. The photogenerated electron-hole pairs are then separated by the built-in field produced by the surface states of Ge (Figure 1c, top panel). The photogenerated holes are pushed to the surface while the electrons move oppositely. Due to the diffusion effect, the photogenerated carriers move continuously to the graphene channel, as shown in Figure 1b. Since the built-in electric field of the graphene/Ge heterojunction points to graphene (Figure 1c, bottom panel), the photogenerated holes will eventually be injected into graphene driven by the built-in field [27,28] and generate a photocurrent signal. Figure 1d shows the simulated electric field distributions of the graphene/Ge PSD and the control device without graphene. It can be seen that by combining graphene and Ge, a strong interface electric field will be formed, which can effectively enhance the collection and injection of photogenerated carriers. Furthermore, benefitting from the high carrier mobility of graphene, the holes injected into graphene can circulate multiple times through the channel, and thus bring in a high photoconductive gain [29].

## 3. Experiments Section

### 3.1. Device Fabrication

Figure 2a shows the fabrication process of the graphene/Ge photoconductive PSD. Firstly, the intrinsic Ge substrate (resistivity 100 Ω·cm) was cleaned by soaking in DI water, acetone, and ethanol in an ultrasonic cleaning machine to remove dust and organic contamination. After that, the double-layer photoresists (Lor5A and S1805) were successively spin-coated onto the Ge substrate. After photolithography, the samples were developed by AZ300, and the electrode patterns on the four corners of the Ge substrate were obtained. Next, 50 nm Au film was deposited onto the Ge substrate by using magnetron sputtering. By immersing the Ge substrate in acetone to dissolve the unexposed photoresists, the electrodes were obtained through the lift-off process. Subsequently, the monolayer graphene grown on Cu foils by chemical vapor deposition was transferred onto the Ge substrate through the polymethyl methacrylate (PMMA) assisted wet transfer method. The Ge substrate covered with graphene/PMMA film was heated on a hot plate at 150 °C for 30 min to make the graphene adhere well to the Ge substrate. After that, the PMMA film was removed by using acetone. To obtain graphene strip patterns, the second double-layer photoresist (Lor5A and S1805) were spin-coated onto the graphene film. The graphene strip patterns were defined by photolithography and development. Subsequently, the uncovered graphene was etched with oxygen plasma for about 2 min. At last, the graphene/Ge photoconductive PSDs were accomplished by submerging the devices in acetone for a few minutes to remove the remaining photoresists.

### 3.2. Device Measurement

Raman measurements were performed under ambient conditions using a Renishaw inVia system with a 532 nm excitation wavelength. During the electric characteristic measurement, the graphene/Ge photoconductive PSD was attached to a printed circuit board and inserted into a socket located at a scanning Galvo System. The printed circuit board was connected to a semiconductor device analyzer Keithley 4200A-SCS for electrical signal read-out. The 1550 nm and 980 nm lasers were focused and vertically irradiated the device through the refraction optical path. The spot diameter of the 1550 nm laser is about 2 μm, and the spot diameter of the 980 nm laser is around 1 μm. The laser and microscope share the same optical path. By exploiting a charge-couple device (CCD) camera, the 980 nm laser light spot (with an overlapping incident position with 1550 nm laser) and the device can be seen in the software, in which the light spot of 980 nm laser is marked with visible color. With the assistance of the software, the alignment of the light spot and device can be realized. To measure the position sensitivity, the scanning system carried the graphene/Ge photoconductive PSD and moved accurately with a given value. To measure the response speed, a signal generator was used to connect the laser and generated a modulated light signal at a frequency of 1 kHz. Then, a SourceMeter (FS-Pro) with a high sampling rate was used to measure the transient response of the device. For the noise measurement, a noise analyzer was connected to the device and recorded the noise current of the device. The measurable value of the noise analyzer is as low as 2 × 10^−28^ A^2^/Hz. When calibrating the power of the laser, a commercial power meter from Thorlabs was utilized. Throughout the measurement, the bias voltage of the device is 1 V.

## 4. Results and Discussion

### 4.1. Device Characterization

The insert of Figure 2b shows the optical microscope image of the graphene/Ge photoconductive PSD. The surface of the device is clean and free of impurity particles. Meanwhile, graphene is in good contact with the electrodes without vacancies. Figure 2b shows the Raman spectrum of graphene. It can be seen that two obvious characteristic bands of graphene are located near 1582 cm^−1^ (G band) and 2700 cm^−1^ (2D band), respectively. The intensity of the 2D band is twice as high as that of the G band, indicating that the monolayer graphene used in this work is of good quality. Figure 2c shows the electric characteristics of devices with and without graphene. It can be seen that the I-V curve of the devices maintains a straight line with good linearity, indicating a good ohmic contact between graphene and the electrodes. Although the introduction of graphene increases the dark current, the detection distance of the device to the light spot position is also improved.

### 4.2. Position-Sensitive Characteristics

Figure 3a shows the position sensitivity of the graphene/Ge photoconductive PSD. It can be seen that the photocurrent of the device decreases as the spot is far away from the graphene channel, which can be attributed to the recombination of photogenerated carriers in the diffusion process. The introduction of graphene greatly enhances the interface electric field, so that the photogenerated carriers can be effectively collected and injected into graphene. Even if the light spot is up to 3000 μm away from the graphene channel, the device still exhibits a photocurrent of 300 nA, which implies that the device can be used for large working areas.

To prove the improvement effect of introducing graphene, the detection distances of the devices with and without graphene are compared. As shown in Figure 3b, the photocurrents of the graphene-Ge device are nearly 3 times higher than those of the pure Ge device, due to the enhanced interface electric field. Figure 3c shows the photocurrent as a function of position at different power, in which it can be seen that the device still exhibits an obvious position-sensitive response when the incident optical power is 1 mW, indicating that the device has a good capability for weak light detection. A 980 nm laser was also used to characterize the position sensitivity, as shown in Figure 3d, which also shows a similar trend and a long distance of 3000 μm. In view of the concealment, information safety, and human eye friendliness of near-infrared light, the excellent near-infrared position sensitivity of the graphene/Ge photoconductive PSD is of great value [30].

### 4.3. Photoresponse Performance

In addition to the position sensitivity characteristics, the photoresponse performance of the graphene/Ge photoconductive PSD was also measured. As shown in Figure 4a, the net photocurrent of the device increases with the increase of light power density and exhibits a good linear relationship. The net photocurrent (*I_net_*) is defined by
*I**_net_* = *I**_p_* − *I**_dark_*(1)
where *I_p_* is the photocurrent and *I_dark_* is the dark current. The noise current of the device is shown in Figure 4b, and it can be seen that the noise current is less than 10^−16^ A^2^/Hz, which provides a strong guarantee for weak light detection and a large working area. Figure 4c shows the responsivity and noise equivalent power of the device as a function of light power density. The responsivity is defined by
*R* = *I_p_*/*P_o_*(2)
where the *P_o_* is the incident light power. Under different light power densities, the average responsivity is about 40 A/W, and the minimum noise equivalent power is 7.15 × 10^−12^ W·Hz^−1/2^. The noise equivalent power is defined by
(3)$$NEP=\frac{I_{n}}{R}$$
where the *I_n_* is the noise current. The high responsivity and low noise equivalent power can be attributed to two factors, namely, high photoconductive gain brought by the high carrier mobility of graphene [31,32,33] and the strong photogenerated carrier collection and injection effect improved by the built-in field [34,35].

Note that the high carrier mobility of graphene and the built-in field of the graphene/Ge heterojunction are vital for the response speed. The former can ensure the rapid transport of carriers through graphene channels, while the latter can accelerate the charge injection. The response speed of the device is shown in Figure 4d, in which the rise time and fall time are merely 3 μs and 1.9 μs, respectively. Such a fast response speed is highly desired for high-speed object motion tracking.

## 5. Conclusions

In summary, PSDs that can detect the position change of the 1550 nm laser spot have been realized by integrating graphene with Ge. The introduction of graphene can produce a strong built-in field at the graphene/Ge heterojunction region, which is helpful for effectively collecting photogenerated carriers and driving these photogenerated carriers to inject into graphene. Thereby, the graphene/Ge photoconductive PSD achieves a fast response and linear near-infrared detection. Moreover, the working distance of the graphene/Ge photoconductive PSD is up to 3000 μm, which is three times higher than that of the bare Ge device. These results provide a promising opportunity for infrared PSDs based on low-dimensional materials and traditional bulk semiconductors.

## Figures and Tables

**Figure 1 nanomaterials-13-00322-f001:**
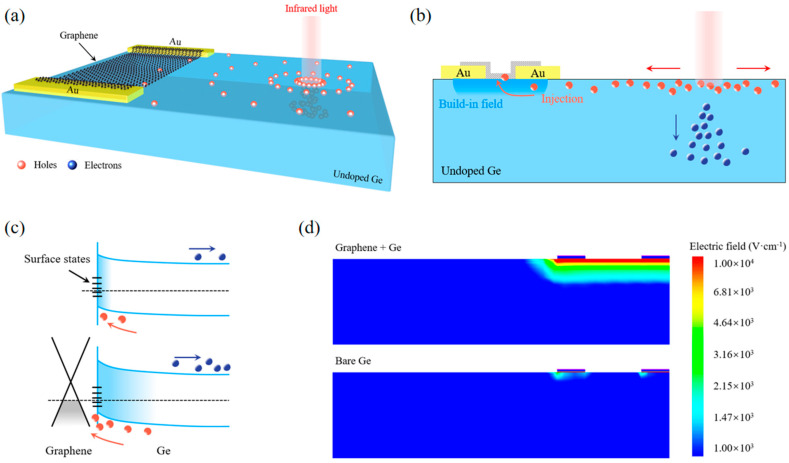
(**a**) Three-dimensional diagram of the graphene/Ge photoconductive PSD. (**b**) Schematic diagram of the diffusion process of photogenerated carriers. (**c**) Energy band diagrams of the devices without and with graphene on Ge substrate. (**d**) Simulation results of electric field distribution for the devices with and without graphene.

**Figure 2 nanomaterials-13-00322-f002:**
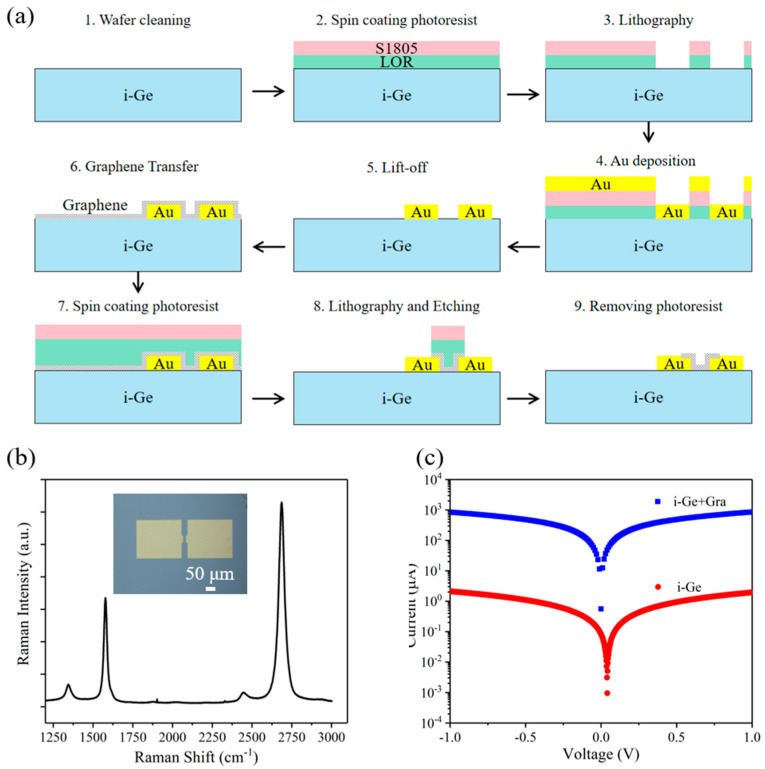
(**a**) Device fabrication process of the graphene/Ge photoconductive PSD. (**b**) Raman spectra of graphene. The inset displays the optical microscope image of the device. (**c**) I–V curves of the graphene/Ge photoconductive PSD and the Ge device.

**Figure 3 nanomaterials-13-00322-f003:**
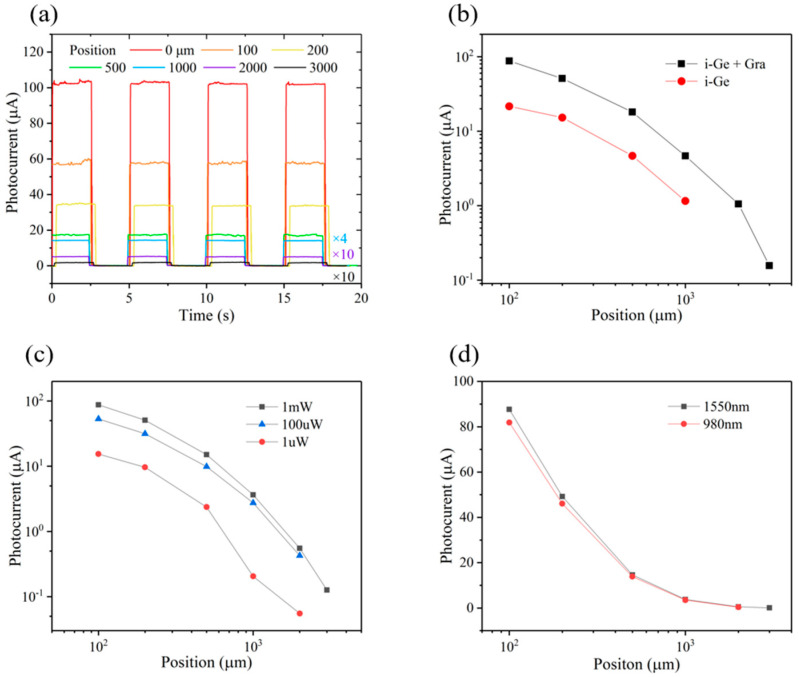
Position-sensitive characteristics of the device. (**a**) Photo-switching characteristics of the device at different positions (1550 nm laser, V_bias_ = 1 V). (**b**) Comparison of the position-sensitive characteristics of the graphene/Ge and pure Ge devices under 1 mW. (**c**) Photocurrent as a function of light spot position under different light power. (**d**) Photocurrent as a function of light spot position under 1550 nm and 980 nm lasers.

**Figure 4 nanomaterials-13-00322-f004:**
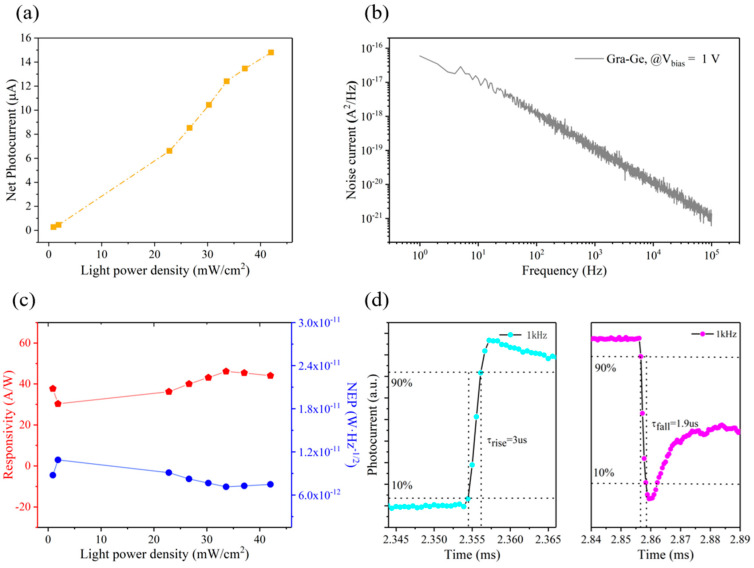
Photoresponse characteristics of the device. (**a**) Net photocurrent as a function of the light power density with the spot scale. (**b**) Noise current. (**c**) Responsivity and noise-equivalent power as a function of light power density. (**d**) Rise time and fall time of the device.

## Data Availability

Not applicable.

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
