# Peer review of "Graphene/Ge Photoconductive Position-Sensitive Detectors Based on the Charge Injection Effect"

_nanomaterials, 2023, doi:10.3390/nano13020322_

Round 1

Reviewer 1 Report

In this manuscript, the authors report position a sensitive detectors (PSDs) based on graphene/Ge heterojunction device, which exhibits amplified signals by unitizing the charge injection effect. A large number of photogenerated carriers diffuse from the incident position of the light spot and subsequently inject into graphene, which ultimately generates a photoresponse with high efficiency. Thereby, the graphene/Ge photoconductive PSD achieves a fast response and linear near infrared detection. The work is very interesting and the structure is experimentally very well characterized. Considering the possible application fields the topic is rather timely and deserves to be considered for publication after minor revisions.

Suggested revisions:

1. What is the spot size of the laser irradiation on the device?

2. Please add the scale bar in the OM image in Fig. 2b.

3. In Fig. 3, the authors measured the position sensitive characteristics of the device, what is the step time of each position measurement? If the step time is too less, there might be some residual heat effect remaining on the device.

4. How do the authors make the alignment process during the position sensitive measurement? The authors mentioned that the lasers were focused and vertically irradiated the device through the refraction optical path, can the authors address more detailed descriptions or draw a brief schematic diagram of the measurement process?

5. There are some typos, such as line 122 on page 6 and line 133 on page 7 “Lor”, line 209 on page 10 “A2/Hz”. In Fig. 4c, the title of the left axis should be “Responsivity” rather than “Responsity”.

6. There are some grammatical errors or some expression might confuse the readers, such as line 191 on page 9 “indicating that the device has a good weak light detection capability”, I would suggest to revise it as “indicating that the device has a good capability for weak light detection” or so.

Author Response

Dear Reviewer,

Thank you for your valuable suggestions. We have amended the paper according to your comments and provided point-by-point responses to the comments. The revisions in the manuscript are highlighted in red color.

Thanks again.

Yours sincerely,

Jun Wu (On behalf of all authors)

Reviewer 2 Report

See attached.

Author Response

(The authors gave the same response as above.)

Round 2

Reviewer 2 Report

Dear authors,

thank you for your exhaustive reply.